# Video-Feedback Approach Improves Parental Compliance to Early Behavioral Interventions in Children with Autism Spectrum Disorders during the COVID-19 Pandemic: A Pilot Investigation

**DOI:** 10.3390/children9111710

**Published:** 2022-11-08

**Authors:** Stefania Aiello, Elisa Leonardi, Antonio Cerasa, Rocco Servidio, Francesca Isabella Famà, Cristina Carrozza, Agrippina Campisi, Flavia Marino, Renato Scifo, Sabrina Baieli, Flavio Corpina, Gennaro Tartarisco, David Vagni, Giovanni Pioggia, Liliana Ruta

**Affiliations:** 1Institute for Biomedical Research and Innovation (IRIB), National Research Council of Italy, 98122 Messina, Italy; 2S. Anna Institute and Research in Advanced Neurorehabilitation (RAN), 88900 Crotone, Italy; 3Pharmacotechnology Documentation and Transfer Unit, Preclinical and Translational Pharmacology, Department of Pharmacy, Health Science and Nutrition, University of Calabria, 87036 Arcavacata, Italy; 4Department of Cultures, Education and Society, University of Calabria, 87036 Arcavacata, Italy; 5Centre for Autism Spectrum Disorders, Child Psychiatry Unit, Provincial Health Service of Catania (ASP CT), 95100 Catania, Italy; 6Multi-Specialist Clinical Institute for Orthopedic Trauma Care (COT), 98124 Messina, Italy

**Keywords:** autism, early interventions, tele-assistance, video feedback, COVID-19

## Abstract

In the field of autism intervention, a large amount of evidence has demonstrated that parent-mediated interventions are effective in promoting a child’s learning and parent caring skills. Furthermore, remote delivery treatments are feasible and can represent a promising opportunity to reach families at distance with positive results. Recently, the sudden outbreak of COVID-19 dramatically disrupted intervention services for autism and forced an immediate reorganization of the territory services toward tele-assisted intervention programs, according to professional and local resources. Our study aimed to conduct a retrospective pilot exploratory investigation on parental compliance, participation, and satisfaction in relation to three different telehealth intervention modalities, such as video feedback, live streaming, and psychoeducation, implemented in the context of a public community setting delivering early autism intervention during the COVID-19 emergency. We found that parents who attended video feedback expressed the highest rate of compliance and participation, while parental psychoeducation showed significantly lower compliance and the highest drop-out rate. Regardless of the tele-assistance modality, all the participants expressed satisfaction with the telehealth experience, finding it useful and effective. Potential benefits and advantages of different remote modalities with reference to parent involvement and effectiveness are important aspects to be taken into account and should be further investigated in future studies.

## 1. Introduction

Autism spectrum disorder (ASD) is a complex neurodevelopmental condition characterized by social communication impairments and restricted and repetitive interests and behaviors [1]. Substantial phenotypic heterogeneity in clinical presentation, developmental trajectories, and behavioral profiles are reported and early intervention exerts a pivotal role in maximizing outcome [2,3,4]. Different kinds of early interventions and delivery modalities have been tested and validated [5,6]. Parent-mediated interventions proved the effectiveness of parents as treatment agents, strengthening the active participation of parents during daily routines as well as supporting their role in the generalization and maintenance of their child’s achievements [7,8,9,10].

Furthermore, parents’ involvement in their child’s treatment contributes to the parents’ psychological well-being, decreasing parental distress, improving their coping strategies, and empowering their self-efficacy and adjustment to their child’s daily requests [11,12,13]. In the last decade, telemedicine [14] has enriched parent-mediated early interventions [15]. Several studies, conducted on a variety of treatment models in children and adolescents with ASD, demonstrated that remote interventions are able to promote parent’s learning and caregiving skills, support a reduction of children’s behavioral problems, and contribute to an improvement in autism symptoms’ severity, language, and adaptive skills [16,17,18]. Furthermore, a few studies compared tele-assisted interventions with in-person treatments, suggesting a comparable [19] or even better outcome of the remote program. This is the case of a recent RCT on ABA-based intervention that found a significant decrease in parental distress and children’s challenging behaviors in the telehealth group [20].

Among different types of tested telehealth delivery methods, live-streaming, as well as video-feedback modalities, showed evidence of efficacy [21].

In the live-streaming modality, the parent practices skills with their child during the video-conferencing sessions and the therapist provides real-time guidance and support [22].

In the video-feedback modality, conversely, the interactions between the parent and the child are videotaped asynchronously; the videos are sent to the therapist before the video-conferencing session and discussed during the online session [23,24,25,26,27]. The video-feedback review moment represents an opportunity for the parent to observe themselves “from the outside” and to reflect on their behaviors and the child’s responses.

The video-feedback approach has been proven to be effective in promoting positive parenting and child development in different at-risk children populations and clinical settings, including children with behavioral problems [28,29], neurodevelopmental conditions [30], and ASD.

Very recently, the sudden outbreaks of the COVID-19 pandemic created a dramatic and sudden disruption of the ongoing treatments. Despite the highly stressful psychological and family burden [31,32,33], the situation forced community services worldwide to promptly switch to telehealth services to ensure a continuum of care and daily-life stimulation for ASD children. Most families, in fact, declared greater than usual difficulties in managing daily tasks and a worsening of their children’s inadequate behaviors [34,35]. The high level of parental psychological distress became a major challenge for ASD families, that in turn affected their supportiveness towards the child [36,37]. A recent survey conducted in the United States (US) on over 3000 parents/caregivers of children with ASD reported that many families did not receive any intervention, while those services that switched to a telehealth format provided limited support and showed benefits, especially for preschool children [38].

Similarly to the US and many other countries, Italy had to face cultural and technological barriers to remote interventions [39]. Although many autism community centers did their best to adapt interventions for online platforms and somehow continued to deliver modified programs to families, most of the adaptations were unsystematic and lacked clear policies or central guidance. Different types of remote intervention, with or without the ASD child’s synchronous involvement, were used, depending on the therapist’s experience, attitude, and family choices. However, to the authors’ knowledge, only one Italian study reported a systematic analysis of the effectiveness of a remote ABA-based intervention during the COVID-19 pandemic, showing a significant improvement in the targeted behavioral problems, as well as a reduction of parental psychological distress [40]. Furthermore, socio-demographic and parental factors related to telehealth interventions’ compliance and efficacy have not been studied.

Based on this framework, we aimed to explore, retrospectively, parental compliance, participation, and satisfaction in relation to three different telehealth intervention modalities, such as live streaming, video feedback, and psychoeducation, implemented in the context of a public community setting delivering early autism intervention during the COVID-19 lockdown.

Based on the suddenness of the COVID-19 pandemic and the urgency of an immediate reorganization of the territory service according to the professional and immediate local resources, we conducted a post-hoc analysis and a comparison of all the demographic, clinical, and COVID-19-related environmental factors that we were able to gather and which could have influenced the parental compliance with different types of remote behavioral intervention.

## 2. Methods

### 2.1. Participants

The study included a cohort of families of young children with ASD who, at the time of the COVID-19 outbreak, were undergoing an early naturalistic behavioral intervention at the public territory service of the Child Psychiatry Unit of the Provincial Health Service (ASP-CT) in Catania, Italy. All the children (*n* = 31) had received a clinical diagnosis of ASD according to the DSM-5 criteria made by a multidisciplinary team (a child neuropsychiatrist, a developmental psychologist, a speech–language therapist, and an educator). DSM-5 severity levels for ASD, related to the social communication domain, are reported in Table 1. Furthermore, a diagnostic assessment was available for approximately half of the sample (57%) and included the ADOS-2 (ADOS-2 classification of autism in 73% of children and ASD in the remaining 27% of the sample) and the Griffiths Mental Development Scale (total developmental quotient (DQ): M = 61 DS = 22). An estimate of the language level was also extracted from the scores at item A1 of the ADOS-2 and reflected the presence of and a number of words spontaneously pronounced.

When the COVID-19 pandemic started, children were receiving in-person 1:1 intervention at the autism clinic, with a frequency of three times a week (approximately 45 min each session). An individualized objective plan, based on the ESDM curriculum checklist was implemented with each child. The service treatment team included qualified and experienced professionals (one speech–language therapist, one occupational therapist, and two educators) with a background in behavioral principles and developmental psychology. Furthermore, all the therapists had received specific training on the Early Start Denver Model (ESDM). They had all attended an introductory and advanced workshop on-site and were undergoing monthly based supervision either in person or remotely with the last author (L.R.), who is a certified ESDM trainer.

### 2.2. Ethics

The study was conducted according to the guidelines of the Declaration of Helsinki and approved by the Ethics Committees of CNR (ethical clearance, 1 August 2018) and ASP-CT (Prot. N. 498), respectively, and all the caregivers signed informed consent to participate in the study.

### 2.3. Procedure

The remote intervention program lasted from the first week of March to the end of June 2020 for a total of 24 sessions, twice a week for approximately 60 min each session. Telehealth interventions were delivered by the same therapists who were already conducting the in-person therapies. Being a fluent Italian speaker was applied as an inclusion criterion to participate in the survey. Exclusion criteria included: (1) genetic factors, (2) epilepsy, and (3) severe sensory and motor deficits. All children in the sample had received a CHG-array screening and none of them had reported genetic conditions associated with autism (e.g., fragile X syndrome and Down syndrome), nor epileptic encephalopathy with onset in infancy, or significant sensory or motor impairment (e.g., vision or hearing impairment, cerebral palsy). Two families were excluded from the study because the parents were foreigners and not fluent in Italian. No children reported other excluding conditions.

Three different telehealth modalities were used, as follows: (i) live streaming (52%), (ii) video feedback (29%), and (iii) parental psychoeducation (19%).

In the **live-streaming** condition, a real-time interaction between the parent and the child occurred during the online sessions and the therapist coached parents directly via video conference; the **video-feedback** modality implied that the parent would record videos of parent–child interaction asynchronously and send the video examples to the therapist before the online session. The videos were then reviewed and discussed during the session with the therapist. Finally, **parental psychoeducation** was conducted as frontal training for the parents without the child involved. Regardless of the telehealth modality, all the therapists kept following the child’s personalized treatment plan and discussed parental behavioral strategies according to the ESDM principles. Although none of the therapists had formal training on ESDM parent coaching, all the therapists in the team had previously read the ESDM manual for parents “An Early Start for Your Child with Autism” [41]. Furthermore, two therapists (one speech–language therapist, and one educator) attended seminars on the video-feedback approach in relation to the pediatric autism communication therapy (PACT) early intervention model.

The families were assigned to one of the three different telehealth modalities based on the therapist’s choice and experience, discussed with the family.

The video-feedback group was assigned to the therapists as follows: five families implemented remote intervention with the speech and language therapist and four families with the educator; in the live-streaming group, two families were assigned to the speech and language therapist and three families to one of the educators, and seven families to the other educator and four families to the occupational therapist; for parental psychoeducation, three families were treated by the occupational therapist, two families by the educator, and one family by the speech and language therapist. Telehealth sessions were conducted using one of the available social communication platforms, such as Skype, WhatsApp, Zoom, and Google Meet, according to the family preference.

### 2.4. Qualitative Analysis

At the end of the lockdown (July 2020), data about parental participation in telehealth protocols—in terms of the number of attended sessions—and modalities were collected through the central database located at IRIB-CNR, Messina. A dedicated survey was then conducted via an ad hoc questionnaire administered to the participants. The survey was a semi-structured interview administered within two weeks from the end of the lock down via phone interview. Each interview lasted approximately 45 min. The survey was conducted with the mothers being the only parent involved in the telehealth sessions.

Two sets of data were gathered; one set related to demographic, environmental, and COVID-related variables, and the second set assessed parental, personal judgment about telehealth.

In the first set, we investigated variables that can interfere with telehealth compliance: (1) **COVID-care burden**; (2) **worries** (COVID-related emotional distress); (3) child **sleep** disturbance, and (4) child **feeding**/evacuation disturbance. Each domain included different questions where either dichotomous (yes/no) or Likert-type answers were attributed. The COVID-care burden domain included: number of children, other children with neurodevelopmental/other medical conditions, other family members to care for, and working or smart-working during the lockdown. Worries domain included: subjective psychological distress, fear of contagion, fear of job loss, concerns about child’s treatment interruption, daily routine changes, and personal life-style changes. The sleep domain included: child difficulties with falling asleep, child nocturnal awakenings, melatonin or other drug needs, and daily naps. The feeding domain included: feeding restriction and selectivity, and constipation. Each value was summed up for each domain to generate a total score.

The second set of answers was related to parents’ telehealth judgments. We investigated parents’ ratings about **telehealth utility** (dichotomous) and **telehealth satisfaction**, in terms of support and utility with their child’s caring (five-point Likert-type scale ranging from one, “useless”, to five, “very useful”). A Likert-type scale was also used to assess **telehealth efficacy**, which was obtained by summing ratings related to perceived usefulness in managing difficult behaviors, receiving emotional support, and learning about their child. Finally, two questions asked about the willingness of continuing with telehealth after the lockdown and the reasons. The survey is included in the Appendix A.

### 2.5. Statistical Analysis and Quantitative Outcome Variables

Statistical analyses were performed using the Statistical Package for Social Sciences software–SPSS (version 26.0, Chicago, IL, USA). The main outcome variables were participation and compliance to the treatment. Participation was calculated as the number of sessions performed before either drop-out or the end of the protocol. Compliance to treatment was calculated as the percentage of sessions (24 maximum) attended by at least one parent and used in quantitative statistical analysis. For instance, if a family participated in all sessions until the 12th and then decided to end the therapy, their participation would be 12 but their compliance would be 50%.

We used a between-group design and performed two one-way analyses of variance (ANOVA), considering the three telehealth modalities as independent variables and compliance and participation as the outcome variables. Levene’s test for equality of variances was used to assess the homogeneity of variance.

## 3. Results

Demographic and clinical characteristics of the three groups of ASD parents are shown in Table 1. The three intervention groups were matched in terms of children’s age, as well as main demographic and clinical characteristics, except for the educational level of mothers, which is greater in the tele-intervention group 1.

Levene’s test for equality of variances was not significant for all the demographic variables, indicating homogeneity of variance across the telehealth modalities (reported in Table 1). The one-way ANOVA indicated a main effect of telehealth modality on both participation (F (2,28) = 21.53, *p* < 0.001) and compliance to treatment (F (2,28) = 6.59, *p* = 0.005). As for participation, pairwise comparisons showed that the-video feedback modality had higher ratings (M = 22.33, SD = 2.55) compared with both live streaming (M = 17.81, SD = 5.07, t (23) = 2.49, *p* = 0.21) and parental psychoeducation (M = 8.00, SD = 3.16, t (13) = 9.27, *p* < 0.001), and live streaming, in turn, displayed higher ratings compared with parental psychoeducation (*t* (20) = 3.54, *p* = 0.002). Similarly, the video-feedback modality showed the highest rate of compliance to treatment (M = 0.93, SD = 0.10) compared with live streaming (M = 0.74, SD = 0.21, *t* (23) = 2.49, *p* = 0.21) and psychoeducation (M = 0.51, SD = 0.35, *t* (5.63) = 2.89, *p* = 0.03), while there was no significant difference in parental compliance comparing live streaming and psychoeducation (*t* (20) = 1.93, *p* = 0.068). Only three families dropped out before the end of the treatment, and they were all in the psychoeducation group (50% of their group total population). A total of 90% of participants judged the experience useful; most participants were enough or very satisfied (M = 4.67, SD = 0.70 for video-feedback modality; M = 4.27, SD = 1.17 for synchronous live interaction; and M = 3.50, SD = 1.23 for parental psychoeducation) and evaluated telehealth parent training effective (M = 4.07, maximum = 12, SD = 1.03). There were no statistically significant differences among different modalities for satisfaction (F (2,27) = 2.19, *p* = 0.13), perceived utility (F (2,27) = 0.22, *p* = 0.80), and perceived efficacy (F (2,27) = 0.32, *p* = 0.78).

Although judging the telehealth experience as useful and effective in consideration of the abrupt COVID-related confinement, 90% of participants would not continue with telehealth only, 78% of them because they thought that in-presence intervention was more effective and 22% of them because they had difficulties with the technologies. The 10% of parents who expressed their willingness to continue with telehealth emphasized the valuable chance to foster a continuum of care during daily routines. The option of a blended model (telehealth and in presence) was also investigated. One third (33%) of parents were in favor of a blended model: 70% of them thought that a blended model would maximize intervention effectiveness, 20% positively evaluated the opportunity of receiving virtual parent coaching from the comfort of their home, and 10% appreciated schedule flexibility allowed by telehealth. Two thirds (67%) of parents who were skeptical about a blended solution reported the following reasons: 85% of them thought that telehealth did not add any more significant gain to in-person treatment and 15% expressed difficulties with technologies.

## 4. Discussion

Parent compliance with interventions is pivotal for achieving treatment goals [42,43]. With COVID-19, the public health emergency forced health-care services to switch to telehealth as a necessary tool for ensuring a continuum of care. Remote parent coaching represented an opportunity for supporting families with young children with special needs [31] and autism [44]. Studies about telehealth in the autism field, with reference to assessment or treatment programs, were already ongoing before the pandemic, but mostly represented circumscribed evidence in rigorous experimental and research settings [45,46]. The sudden and unpredicted disruption of in-person therapy and the necessary transition to virtual services, as the only alternative to continue rehabilitation programs, has dramatically shifted the debate about placing remote modalities side by side with face-to-face programs at the ASD community level. However, while anecdotal evidence about advantages and weaknesses of different virtual therapy modalities is reported, results from well-designed studies are still minimal.

According to the de Nocker and Toolan review [47] only a few studies on group designs have been implemented, mostly presenting case studies instead. To the author’s knowledge, only one recent study reported on parents’ satisfaction with teleassistance in children with neurodevelopmental disorders during the COVID-19 pandemic in Italy. Results from the online survey indicated that tele-rehabilitation was well accepted by the majority of families [48]. In our study, we systematically tested, retrospectively, parental compliance, participation, and satisfaction on three different telehealth programs, respectively, video feedback, live-streaming interaction, and parental psychoeducation in a cohort of very young children with autism who, at the beginning of COVID-19 pandemic, were undergoing a naturalistic behavioral intervention.

According to our pilot investigation, parents who attended video-feedback therapy sessions expressed the highest rate of compliance and participation, while parental psychoeducation showed significantly lower compliance and the highest drop-out rate. It is possible to hypothesize that, considering the difficulties related to the COVID-19 lockdown, video feedback allowed parents greater flexibility, a moment of guided reflection without having to manage their child’s behavior or other external contingencies and reflected ultimately better adherence and satisfaction.

Our findings are also consistent with previous evidence reporting that video feedback is an effective and well-accepted strategy to coach parents of young children with autism in various intervention models [25,26,27].

With reference to the live-streaming modality, early naturalistic developmental behavioral interventions, such as the ESDM, have contributed to testing the feasibility and efficacy of supporting parents’ learning and positive caring skills using a reflective style and practice [19,20]. The strength of this approach is that the parent has the opportunity to interact with the child in the virtual presence of the therapist, reflect on what happened, and try again, at the moment, to put into practice the discussed strategies. It is likely that, in the COVID-19 context, having to arrange a real-time interaction with the child in the virtual presence of the therapist was perceived as an extra challenge and impacted the slightly lower parental compliance to this remote intervention modality.

Interestingly, regardless of the tele-assistance modality, all the participants expressed satisfaction with the telehealth experience, finding it useful and effective.

However, despite these positive judgments, a significant minority of parents would have continued with telehealth (either alone or in a blended model), possibly suggesting that parents circumscribed benefits to the contingent situation of confinement in the hope that a return to normal life would maintain their previous treatment routine. Furthermore, it is worth mentioning that a minority of parents found that technological barriers outweighed the benefits.

Among the parents who would have willingly participated in a blended model, many of them thought that remote parent coaching was a unique opportunity to promote a continuum of care during their child’s daily routines and to foster a generalization of skills in an ecological context. Flexibility in time schedule and geographical distances were also valued.

### Limitations

The study wants to pose itself as a pilot exploratory investigation conducted in a small sample of parents who found themselves forced to switch to an unfamiliar intervention modality in a specific, abrupt, health-related emergency. Hence, although valuing the effort to collect information on the impact of different telehealth modalities on parental compliance, participation, and satisfaction, retrospectively, several limitations have to be considered.

Firstly, the sample size of the study is very limited and, hence, a generalization of results has to be viewed with great caution. Another limitation of the study is represented by its retrospective nature, which did not allow us to randomize participants and therapists. However, overall, no significant group difference was found in the demographic variables, thus limiting the presence of possible confounding variables. Only the mothers’ education was slightly higher in the video feedback group, and it is worth mentioning that it could have impacted their motivation to participate. Furthermore, the clinical assessment was available for about half of the children, and it did not allow any kind of analysis on possible correlations between telehealth modality and children’s improvement. Finally, it is important to note that our study was focused on young children undergoing an early naturalistic developmental behavioral intervention, based on the ESDM, and it is not straightforward to extend the results to other age ranges, and intervention modalities that should be addressed in the future research.

## 5. Conclusions

Our study aimed to conduct a retrospective pilot exploratory investigation on parental compliance, participation, and satisfaction in relation to three different telehealth intervention modalities, such as video feedback, live streaming, and psychoeducation, implemented in the context of a public community setting delivering early autism intervention during the COVID-19 emergency.

We found that parents who attended video feedback expressed the highest rate of compliance and participation, while parental psychoeducation showed significantly lower compliance and the highest drop-out rate.

It is possible to hypothesize that, taking into account the specific COVID-19-related environmental situation and parental emotional struggle, the video-feedback approach allowed parents greater flexibility and the opportunity to observe themselves “from the outside” through the video review, which contributed to maximizing their engagement and participation. It is also possible that the mothers’ higher level of education in the video feedback group could have made them more comfortable with observing and discussing their own behavior, consequently affecting their participation rate. Regardless of the teleassistance modality, all the participants expressed satisfaction with the telehealth experience, finding it useful and effective. Besides the limitations, our study aimed to contribute to the emerging literature on how the COVID-19 pandemic has accelerated telehealth support to parents of children with ASD in the Italian context, and how different telehealth modalities may affect parental compliance and learning outcomes. With this regard, future studies should take into consideration if parental educational level affects parents’ motivation and compliance in parent coaching programs. Finally, all these implications should be translated into a more general framework, with different ASD clinical phenotypes, across different ages and different treatment models.

## Figures and Tables

**Table 1 children-09-01710-t001:** Demographic and psychological characteristics of ASD parental groups enrolled in the tele-intervention protocols.

Variables	Tele-Intervention 1: *Parent Coaching through Video Feedback* (*n* = 9)	Tele-Intervention 2: *Synchronous Model through Parent–Child Live-Streaming Interaction* (*n* = 16)	Tele-Intervention 3: *Parental Psychoeducation* (*n* = 6)	*F*-Values	*p-Level*	*Levene Statistic*	*p-Level*
Age Mother (years)	36.2 ± 3.9	37.2 ± 5.8	35 ± 5.1	0.42	0.66	0.52	0.60
Age Father (years)	38.8 ± 4.9	40.9 ± 7.1	37.5 ± 6.8	0.7	0.51	0.74	0.49
Age Child (months)	42.2 ± 14.2	41.1 ± 10.1	40.3 ± 10.8	0.06	0.94	1.83	0.18
Gender (%f) **	33%	19%	33%	0.40	0.67	1.08	0.59
Educational Level Mother [Years]	13 [8,9,10,11,12,13,14,15,16,17,18]	10.5 [8,9,10,11,12,13,14,15,16,17,18]	8 [8,9,10,11,12,13]	3.65	0.04 *	0.82	0.45
Educational Level Father [Years]	13 [13,14,15,16,17,18]	13 [8,9,10,11,12,13,14,15,16,17,18]	10.5 [8,9,10,11,12,13]	2.17	0.13	2.98	0.07
DSM-5 Social-Communication Domain Level	Level 1 = 11%	Level 1 = 19%	Level 1 = 33%	0.57	0.77	2.73	0.08
	Level 2 = 78%	Level 2 = 44%	Level 2 = 33%				
	Level 3 = 11%	Level 3 = 37%	Level 3 = 33%				
Language Level	1.56	1.56	1.5	0.14	0.99	1.63	0.21
Other Medical Conditions	22%	27%	17%	0.80	0.92	0.36	0.70
N. of Children in the Family	1.89	2.00	2.33	0.40	0.68	2.00	0.15
Parental Burden	1.67	1.40	1.67	0.12	0.89	0.80	0.46
Worries	1.33	0.13	0.00	1.22	0.31	0.90	0.91
Sleep Problems	4.22	3.47	2.83	0.70	0.51	0.32	0.73
Feeding Problems	1.22	2.07	1.00	2.04	0.15	0.78	0.47

Note. * *p* < 0.05; ** Kruskal–Wallis Test was performed instead of Levene’s Test.

## Data Availability

The datasets generated during the current study are available from the corresponding author on reasonable request.

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
