# Peer review of "Video-Feedback Approach Improves Parental Compliance to Early Behavioral Interventions in Children with Autism Spectrum Disorders during the COVID-19 Pandemic: A Pilot Investigation"

_children, 2022, doi:10.3390/children9111710_

Round 1
Reviewer 1 Report
I think this is an important paper. The topic is relevant and useful, but it is unclear what the author (s) are hoping parents of ASD or centers to do with the results. In other words, little to no implications/recommendations are provided within the text. The author (s) also need to state their conclusions clearly.
Author Response
Q1) Broadly speaking, an extensive English review is recommended (with greater attention to introduction and discussion). I have highlighted only major grammar errors. I suggest a review from a native English speaker.
R1) We thank the Reviewer for the suggestion. The paper has been carefully revised by a professional language editing service to improve grammar and readability. All the suggestions reported have been addressed.
Please find here some suggestions:
Q1a) Results demonstrated that video-feedback was the telehealth intervention most attended
and showed the highest values for compliance to treatment, compared to live-streaming and parental psychoeducation.
R1a) Done
This part should be rephrased.
Q1b) which affects social interactions, adaptive functioning, and learning, mainly due to social communication impairments, restricted and repetitive interests, and behaviors as well as atypical responses to environmental sensory stimuli
it would be better to write “which could affect”.
However, I suggest removing this consideration and to focus on diagnostic description.
R1b) We have followed the Reviewer’s suggestions and reported DSM-5 autism diagnostic criteria.
Q2) Furthermore, ASD is note “due to” all the symptoms described.
- ASD represents a lifelong condition
Despite usually correct, this sentence does not take into account the small number of people who lost diagnosis later in life. I suggest to rephrase this part.
R2) We agree with the Reviewer’s suggestion and have rephrased as follows:
"Substantial phenotypic heterogeneity in clinical presentation, developmental trajectories, and behavioral profiles are reported."
Q3) - Furthermore, telehealth ensured outcome gains and parent fidelity comparable to the in-person modality
I suggest to spend more words on how telehealth ensured outcome gains comparable to in-modality (i.e. this sentence is true for all autism treatment?)
R3) We thank the Reviewer for raising this point. We realized that the sentence was misleading and have rephrased as follows:
"Several studies, conducted on a variety of treatment models in children and adolescents with ASD, demonstrated that remote interventions are able to promote parent’s learning and caregiving skills, support a reduction of children's behavioral problems, and contribute to an improvement in autism symptoms severity, language, and adaptive skills. Furthermore, a few studies compared tele-assisted interventions with in-person treatments, suggesting a comparable or even better outcome of the remote program. This is the case of a recent RCT on ABA-based intervention that found a significant decrease in parental distress and children’s challenging behaviors in the telehealth group."
Q4) - Video examples are then chosen by the therapist and discussed during the online session
Not sure that the choice of the video examples (maybe video samples?) is up to the therapist in all therapy (i.e. PACT).
R4) We understand the Reviewer’s comment and have reworded as follows:
"In the video-feedback modality, conversely, the interactions between the parent and the child are videotaped asynchronously, and videos are sent to the therapist before the video-conferencing session and discussed during the online session".
Q5)- The following inclusion criteria were 105 applied to participate in the survey: (1) being a fluent Italian speaker; (2) being a biological 106 parent.
I suggest to move this part in procedure, as well as to add a sentence on exclusion criteria.
R5) As suggested by the Reviewer, we have moved this part in the procedure session and have added a sentence on exclusion criteria as follows:
"Exclusion criteria included: (1) genetic factors, (2) epilepsy and (3) severe sensory and motor deficits. "
Q6) - All the children had a clinical diagnosis of ASD according to the DSM-5 criteria, made by an experienced child neuropsychiatrist and supported by the ADOS-2
ASD diagnosis should result from a multi professional assessment (Child Psychiatrist, Psychologist, Speech Language Therapist and so on). Please provide information about the presence of different specialists. If not, add this part in limitations section of this research
R6) We thank the Reviewer for highlighting this point. Indeed a multidisciplinary team was involved in the diagnostic assessment. We reworded as follows:
"All the children (n=33) had received a clinical diagnosis of ASD according to the DSM-5 criteria, made by a multidisciplinary team (a child neuropsychiatrist, a developmental psychologist, a speech-language therapist, an educator, and a social worker)."
Q7) Furthermore, please provide additional information on presence/absence of genetic condition and or medical illness like epilepsy (furthermore these features were considered in inclusion/exclusion criteria?). If these data are not available, add this part in the limitations section of this research.
R7) We thank the Reviewer for this comment. We had the required information and added it to the manuscript:
"None of the children in the sample had reported genetic conditions associated with autism (e.g. Fragile X syndrome and Down syndrome), nor epileptic encephalopathy with onset in infancy, or significant sensory or motor impairment (e.g.vision or hearing impairment, cerebral palsy). 2 families were excluded from the study because the parents were foreigners and not fluent in Italian. No children reported other excluding conditions."
Q10) Finally, provide the time in which assessments have been administered.
R10) We thank the Reviewer for this comment. We have merged this information with the one requested later on about which parent was involved and who was answering the interview. We have added the missing data as follows:
"A dedicated survey was then conducted via an ad hoc questionnaire administered to the participants. The survey was a semi-structured interview administered within two weeks from the end of the lockdown via phone interview. Each interview lasted about 45 minutes. The survey was conducted with the mothers being that they were the main parent involved in the telehealth sessions."
Q11) A final sample of 31 parents of children was included in the study
Not sure if 31 parents or parents of 31 children. Please specify the number of parents.
Furthermore, specify if the mother and father were both involved or not. If only one parent participated, please explicate the distribution of fathers/mothers in the three groups and make consideration on the possible effect of this on participation/compliance.
R11) We thank the Reviewer for this comment. Mothers were the main parent involved in the telehealth program for all the children. As already pointed out, we have added the missing information in the manuscript as follows:
"The survey was conducted with the mothers being that they were the main parent involved in the telehealth sessions."
Q12) - 2.3. Procedure
Please provide more information about past professional’s training. Professionals involved in this project have delivered video feedback based on some models previously studied? If not, have they developed and shared some procedures or principles? Otherwise, each professional has worked based on their experience? Which was their previous training? (EIBI?ESDM?PACT?)
R12) We thank the Reviewer for this suggestion and we have added the missing pieces of information as following:
"The service treatment team included qualified and experienced professionals (1 speech language therapist, 1 occupational therapist and 2 educators) with a background in behavioral principles and developmental psychology. Furthermore, all the therapists had received specific training on the Early Start Denver Model (ESDM). They had all attended an Introductory and Advanced Workshop on site and were undergoing monthly based supervisions either in person or remotely with the last author (L.R.), who is a certified ESDM trainer."
All the therapists kept following the child’s personalized treatment plan and discussed parental behavioral strategies according to the ESDM principles. Although none of the therapists had formal training on ESDM parent coaching, all the therapists in the team had previously read the ESDM manual for parents “An Early Start for Your Child with Autism” [41]. Furthermore, two therapists (1 speech-language therapist, and 1 educator) attended seminars on the video-feedback approach in relation to the pediatric autism communication therapy (PACT) early intervention model."
Q13) Furthermore, you should provide more information about the three different modalities of intervention (at least principles of intervention and the goals of therapy)
R13) We thank the Reviewer for this suggestion and we have added the missing pieces of information as following:
"When the COVID-19 pandemic started, children were receiving in-person 1:1 intervention at the autism clinic, with a frequency of three times a week (about 45 minutes each session). An individualized objective plan, based on the ESDM curriculum checklist was implemented with each child. Regardless of the telehealth modality, all the therapists kept following the child’s personalized treatment plan and discussed parental behavioral strategies according to the ESDM principles."
Q14) Finally you should describe which group have been treated by which professional (i.e. Intervention 1: 2 families treat by Educators, 5 from SLP and so on)
- The families were assigned to one of the three different telehealth modalities based on the
therapist’s choice and experience, discussed with the family.
Please provide more information about criteria assignment.
R14) We addressed the Reviewer’s comment and added the information in the manuscript:
"The video-feedback group was assigned to the therapists as follows: 5 families implemented remote intervention with the speech and language therapist and 4 families with the educator; in the live-streaming group 2 families were assigned to the speech and language therapist, 3 families to one of the educator; 7 families to the other educator, 4 families to the occupational therapist; for parental psychoeducation, 3 families were treated by the occupational therapist, 2 families by the educator and 1 family by the speech and language therapist."
Q15) - The survey was a semi-structured interview administered to the parents. Who answered? Mother? Father? Both together?
R15) We thank the Reviewer for this comment. As previously stated, mothers answered the survey.
Q16) - Table 1. Number of children for each group is not written (it also should be compared with Levene’s test).
R16) We thank the Reviewer for raising this point. We added missing information in the table and reported Levene's test results.
Q17) DSM-5 Level are different between Social communication and Restricted, repetitive behaviors, so that a child could have a level 2 social communication impairment and level 3 Restricted, repetitive behaviors impairment. Please provide information on how you decide to provide a single level. Furthermore, you should specify if Ados-2 scores have been used to determine the levels.
R17) We thank the Reviewer for raising this point. We have specified the missing pieces of information as follows:
"DSM-5 severity levels for ASD, related to the social communication domain, are reported in Table 1. ADOS-2 was not administered yet to all participants due to Covid-19 restrictions. We have added this information in the limitation paragraph."
Q18) You should specify if age of parents and child are expressed in years or months
R18) We thank the Reviewer for this comment. Parents’ age is expressed in years while children’s age is expressed in months. We have specified it in the table.
Q19) It is not clear how language level is defined.
R19) We thank the Reviewer for raising this point. An estimate of the language level was also extracted from the scores at item A1 of the ADOS-2 and reflected the presence of the number of words spontaneously pronounced.
Q20) N. of children. I guess is the number of sons and daughters in family, but it’s not clear.
R20) We thank the Reviewer for this comment. Indeed we intended the number of sons and daughters in the family. We have now rephrased it as n° of children in the family hoping it is clearer now.
Q21) 90% of participants judged the experience useful; Decide if use words or number to express percentual.
R21) We have modified the word to number, as suggested.
Q22) NDBI strategies. Low frequencies Acronyms should be spelled
R22) We thank the Reviewer for this comment. We have now spelled all acronyms in the text.
Q23)- In-person therapies are still felt by the parents as the main opportunity for their child to learn and improve and represent an important respite moment. Is this a personal opinion? Or based on some evidence/communication?
R23) We thank the Reviewer for this precious remark. We have deleted personal opinions about reasons behind parents’ preferences and rephrased reporting parents’ answers to the interview. Although judging the telehealth experience as useful and effective in consideration of the abrupt covid related confinement, 90% of participants wouldn’t continue with telehealth only, 78% of them because they thought that in-presence intervention was more effective and 22% of them because they had difficulties with the technologies [...]. Two-thirds (67%) of parents who were skeptical about a blended solution reported the following reasons: 85% of them thought that telehealth did not add any more significant gain to the in-person treatment and 15% expressed difficulties with technologies.
Q24) Our findings support video-feedback efficacy as a useful tool to reflect on their own behavior. There is no evidence in this research about this. This is a personal opinion supported by previous research.
R24) We thank the Reviewer for this comment. We have deleted personal opinions and have rephrased as follows:
"Our findings are also in line with previous evidence reporting that video feedback is an effective and well-accepted strategy to coach parents of young children with autism, in various intervention models"
Q25) The main limitation of this study is the small sample size and the inhomogeneity in 266 the allocation among the intervention groups.
You should describe the numbers of the specific groups.
R25) We thank the Reviewer for this comment. We have specified the number of participants for each treatment group in Table 1 and reported the small sample size in the limitation as follows:
"In this study, we sought to perform a pilot exploratory investigation conducted on a small sample of parents who found themselves forced to switch to an unfamiliar intervention modality in a specific abrupt health-related emergency. Hence, although valuing the effort to collect information on the impact of different telehealth modalities on parental compliance, participation and satisfaction, retrospectively, several limitations have to be considered. Firstly, the sample size of the study is very limited and hence generalization of results has to be viewed with great caution. "
Q26) The other main limitation is the lack of information on cognitive/developmental level. If you have this information, even descriptively you should add it.
R26) We thank the Reviewer for raising this point. We have added the missing information and added this point to our limitation paragraph too. We specified as following:
"Furthermore, a diagnostic assessment was available for about half of the sample (57%) and included the ADOS-2 (ADOS-2 classification of Autism in 73% of children and ASD in the remaining 27% of the sample) and the Griffiths Mental Development Scale (Total Developmental Quotient (DQ): M = 61 DS = 22).
Furthermore, the clinical assessment was available for about half of the children and it did not allow any kind of analysis of possible correlations between telehealth modality and children’s improvement.
Q27) - Overall, we provided evidence that video-feedback procedure is highly recommended as the main modality for developing parents’ reflective function and mentalization ability to remotely sustain awareness of the child’s state of mind and intentionality.
This conclusion is too drastic for the small number. Please rephrase this part.
R27) We thank the Reviewer for this comment. We have rephrased our conclusions, deleting drastic statements and adding more conditional sentences.
Q28) Broadly speaking, discussion and conclusion must be less drastic. I suggest the use of conditional sentences given the small numbers of this research and the lack of literature.
R28) discussion and conclusion have been changed accordingly.
Q29) Finally, I suggest to add “a pilot investigation” to the title given the small numbers involved.
R29) Done
Reviewer 2 Report
Thanks for the opportunity to review this manuscript. The topic of this research has clinical and research importance. Broadly speaking, an extensive English review is recommended (with greater attention to introduction and discussion). I have highlighted only major grammar error. I suggest a review from a native English speaker.
Please find here some suggestions:
- Results demonstrated that video-feedback was the telehealth intervention most attended and showed the highest values for compliance to treatment, compared to live-streaming and parental psychoeducation.
This part should be rephrased.
- which affects social interactions, adaptive functioning, and learning, mainly due to social communication impairments, restricted and repetitive interests, and behaviors as well as atypical responses to environmental sensory stimuli
it would be better to write “which could affect”. However, I suggest to remove this consideration and to focus on diagnostic description. Furthermore, ASD is note “due to” all the symptoms described.
- ASD represents a lifelong condition
Despite usually correct, this sentence does not take into account the small number of people who lost diagnosis later in life. I suggest to rephrase this part.
- Among them parent-mediated interventions
Delete among them
- parental satisfaction and a sense of agency
delete “a”
- Furthermore, telehealth ensured outcome gains and parent fidelity comparable to the in-person modality
I suggest to spend more words on how telehealth ensured outcome gains comparable to in-modality (i.e. this sentence is true for all autism treatment?)
- Video examples are then chosen by the therapist and discussed during the online session
Not sure that the choice of the video examples (maybe video samples?) is up to the therapist in all therapy (i.e. PACT).
- The following inclusion criteria were 105 applied to participate in the survey: (1) being a fluent Italian speaker; (2) being a biological 106 parent.
I suggest to move this part in procedure, as well as to add a sentence on exclusion criteria.
- All the children had a clinical diagnosis of ASD according to the DSM-5 criteria, made by an experienced child neuropsychiatrist and supported by the ADOS-2
ASD diagnosis should result from a multi professional assessment (Child Psychiatrist, Psychologist, Speech Language Therapist and so on). Please provide information about the presence of different specialists. If not, add this part in limitations section of this research
Furthermore, please provide additional information on presence/absence of genetic condition and or medical illness like epilepsy (furthermore these features were considered in inclusion/exclusion criteria?). If these data are not available, add this part in limitations section of this research.
Finally, provide the time in which assessments have been administered.
- - A final sample of 31 parents of children was included in the study
Not sure if 31 parents or parents of 31 children. Please specify the number of parents.
Furthermore, specify if mother and father were both involved or not. If only one parent participated, please explicate the distribution of fathers/mothers in the three group and make consideration on the possible effect of this on participation/compliance.
- 2.3. Procedure
Please provide more information about past professional’s training. Professionals involved in this project have delivered video feedback based on some models previous studied? If not, they have developed and shared some procedures or principles? Otherwise, each professional has worked based on their experience? Which was their previous training? (EIBI?ESDM?PACT?)
Furthermore, you should provide more information about the three different modalities of intervention (at least principles of intervention and the goals of therapy)
Finally you should describe which group have been treated by which professional (i.e. Intervention 1: 2 families treat by Educators, 5 from SLP and so on)
- The families were assigned to one of the three different telehealth modalities based on the therapist’s choice and experience, discussed with the family.
Please provide more information about criteria assignment.
- The survey was a semi-structured interview administered to the parents.
Who answered? Mother? Father? Both together?
- Table 1
Number of children for each group is not written (it also should be compared with Levene’s test).
DSM-5 Level are different between Social communication and Restricted, repetitive behaviors, so that a child could have a level 2 social communication impairment and level 3 Restricted, repetitive behaviors impairment. Please provide information on how you decide to provide a single level. Furthermore, you should specify if Ados-2 scores have been use to determine the levels.
You should specify if age of parents and child are expressed in years or months
It is not clear how language level is defined.
N. of children. I guess is the number of sons and daughters in family, but it’s not clear.
- - 90% of participants judged the experience useful;
Decide if use words or number to express percentual.
- - NDBI strategies
Low frequencies Acronyms should be spelled
- In-person therapies are still felt by the parents as the main opportunity for their child to learn and improve and represent an important respite moment
Is this a personal opinion? Or based on some evidence/communication?
- Our findings support video-feedback efficacy as a useful tool to reflect on their own behavior
There in no evidence in this research about this. This is a personal opinion supported by previous research.
- The main limitation of this study is the small sample size and the inhomogeneity in 266 the allocation among the intervention groups.
You should describe the numbers of the specific groups. The other main limitation is the lack of information on cognitive/developmental level. If you have this information, even descriptively you should add it.
- Overall, we provided evidence that video-feedback procedure is highly recommended as the main modality for developing parents’ reflective function and mentalization ability to remotely sustain awareness of the child’s state of mind and intentionality.
This conclusion is too drastic for the small number. Please rephrase this part.
Broadly speaking, discussion and conclusion must be less drastic. I suggest the use of conditional sentences given the small numbers of this research and the lack of literature.
Finally, I suggest to add “a pilot investigation” to the title given the small numbers involved.
Author Response
I think this is an important paper. The topic is relevant and useful, but it is unclear what the author (s) are hoping parents of ASD or centers to do with the results. In other words, little to no implications/recommendations are provided within the text. The author (s) also need to state their conclusions clearly.
REPLY: We would like to thank this Reviewer for this evaluation of our work. In the attached new version we have rephrased and improved our conclusions
Round 2
Reviewer 2 Report
The paper has been significantly improved. However, some parts required further modifications. Please find here some comments:
1. "being a biological parent" is an inclusion criteria.
Why? There are no reason why a "non biological parent" could have not been included. Please provide a rationale for that.
2. "had reported genetic conditions"
I is not clear if all partecipants have received a genetic screening and, thereby, they could exclude the presence of genetic condition? maybe they just don't know if they have genetic condition?
3. All the children (n=33) had received a clinical diagnosis of ASD according to the DSM-5 criteria,
the diagnoses whas based on what? assessment? only observation?
Furthemore, the final sample after applying exclusion criteria is 31 and not 33
Furthemore, wht's the role of social worker in diagnostic process?
4. "The survey was conducted with the mothers being that they were the main parent involved in the telehealth sessions."
Please add in procedure a sentence to explain if only mothers have been involved. If also fathers, or both, have been involved in the parent training session, report the ratio fathers:mothers:both.
5. "[...] manage their child’s behavior or other external contingencies and reflected ultimately better adherence and satisfaction"
Please add some consideration on the role of Educational Level Mother with regards to these results. The significant highest level of education of mother in video-feedback group could have affected the the highest rate of compliance and participation?
Could you make some consideration about the matching between parental level of education and the kind of parent training modality? i.e. maybe parent with highest level of education could be more confortable with discussing on their own behavior by seeing themselves on a video? or what else?
furthermore, you stated that "telehealth modalities based on the therapist’s choice and experience, discussed with the family." in this choice, education level has been considered?
6. Our findings are also in line with
consistent with*
7. " [...] experience, finding it useful and effective"
please add a sentence in conclusion about the role of educational level, and the implication of this for future studies.
Author Response
Reviewer 2
The paper has been significantly improved. However, some parts required further modifications. Please find here some comments:
Q1) "being a biological parent" is an inclusion criteria.
Why? There is no reason why a "non biological parent" could have not been included. Please provide a rationale for that.
R1) We thank the Reviewer for this comment. The study was part of an ongoing study where
“being a biological parent” was a pertinent criterion. Indeed, in this case, it is not relevant and we removed it from the manuscript.
Q2) "had reported genetic conditions"
It is not clear if all participants have received a genetic screening and, thereby, they could exclude the presence of genetic conditions? maybe they just don't know if they have a genetic condition?
R2) We thank the Reviewer for this comment. All participants had received a CGH-Array screening and the results were negative. We added this information to the manuscript.
Q3) All the children (n=33) had received a clinical diagnosis of ASD according to the DSM-5 criteria, the diagnosis was based on what? assessment? only observation?
R3) We thank the Reviewer for this comment. ASD diagnosis was supported for all the children by the administration of ADOS 2. Furthermore, as already reported in the limitations, about half of the children had repeated the ADOS 2 (and the Griffiths Mental Developmental Scale) immediately before the lock-down, in the context of an ongoing longitudinal study on the efficacy of the ESDM.
Q4) Furthemore, the final sample after applying exclusion criteria is 31 and not 33
R4) Corrected, thanks for spotting it.
Q5) Furthemore, what's the role of social workers in the diagnostic process?
R5) We thank the Reviewer for raising this point. In the context of the clinical territory service, the social worker was part of the multidisciplinary team and had a role in ensuring a coordinated approach between the team and the family and case management. The social worker was not directly involved in the administration of the assessments and to be more precise we have removed the social worker from the manuscript.
Q6) "The survey was conducted with the mothers being that they were the main parent involved in the telehealth sessions."
Please add in procedure a sentence to explain if only mothers have been involved. If also fathers, or both, have been involved in the parent training session, report the ratio fathers:mothers:both.
R6) We thank the Reviewer for this comment. Only mothers participated in the teleassistance sessions and survey. We substituted “the main” with “the only”
Q7) "[...] manage their child’s behavior or other external contingencies and reflected ultimately better adherence and satisfaction"
Please add some consideration on the role of Educational Level Mother with regards to these results. The significant highest level of education of mothers in video-feedback group could have affected the highest rate of compliance and participation?
Could you make some consideration about the matching between parental level of education and the kind of parent training modality? i.e. maybe parents with the highest level of education could be more comfortable with discussing their own behavior by seeing themselves on a video? or what else?
furthermore, you stated that "telehealth modalities based on the therapist’s choice and experience, discussed with the family." In this choice, education level has been considered?
R7) We thank the Reviewer for this suggestion. We agree that the level of education could have impacted mothers’ motivation to participate. We haven’t considered the educational level in the choice of telehealth modalities and we didn’t discuss this aspect in the manuscript broadly. With this regard, we now mentioned it in the limitations and conclusions paragraphs.
Q8) Our findings are also in line with
consistent with*
R8) Thanks, we have changed it according to your suggestion.
Q9) " [...] experience, finding it useful and effective"
please add a sentence in conclusion about the role of educational level, and the implication of this for future studies.
R9) We thank the Reviewer for this suggestion and have added a sentence in the conclusion.